# Combined PTPN11 and MYBPC3 Gene Mutations in an Adult Patient with Noonan Syndrome and Hypertrophic Cardiomyopathy

**DOI:** 10.3390/genes11080947

**Published:** 2020-08-17

**Authors:** Martina Caiazza, Marta Rubino, Emanuele Monda, Annalisa Passariello, Adelaide Fusco, Annapaola Cirillo, Augusto Esposito, Anna Pierno, Federica De Fazio, Roberta Pacileo, Eloisa Evangelista, Giuseppe Pacileo, Maria Giovanna Russo, Giuseppe Limongelli

**Affiliations:** 1Inherited and Rare Cardiovascular Diseases, Department of Translational Medical Sciences, University of Campania “Luigi Vanvitelli”, Monaldi Hospital, 80131 Naples, Italy; martina.caiazza@yahoo.it (M.C.); rubinomarta@libero.it (M.R.); emanuelemonda@me.com (E.M.); annalisapassariello75@gmail.com (A.P.); adelaidefusco@hotmail.it (A.F.); cirilloannapaola@gmail.com (A.C.); augustoesposito1990@gmail.com (A.E.); federica.defaz@gmail.com (F.D.F.); pacileoroberta@gmail.com (R.P.); mariagiovanna.russo@unicampania.it (M.G.R.); 2Department of Physiology of Nutrition, University of Naples “Federico II”, 80138 Naples, Italy; anna.pierno@studenti.unina.it; 3Ames Genetic Laboratory, 80138 Naples, Italy; eloeva@hotmail.it; 4Cardiomyopathies and Heart Failure Department, Monaldi Hospital, 80138 Naples, Italy; gpacileo58@gmail.com; 5Institute of Cardiovascular Sciences, University College of London and St. Bartholomew’s Hospital, London WC1E 6DD, UK

**Keywords:** Noonan syndrome, hypertrophic cardiomyopathy, double mutations

## Abstract

In this report, an atypical case of Noonan syndrome (NS) associated with sarcomeric hypertrophic cardiomyopathy (HCM) in a 33-year-old patient was described. Genetic testing revealed two different disease-causing mutations: a mutation in the PTPN11 gene, explaining NS, and a mutation in the MYBPC3 gene, known to be associated with HCM. This case exemplifies the challenge in achieving a definite etiological diagnosis in patients with HCM and the need to exclude other diseases mimicking this condition (genocopies or phenocopies). Compound heterozygous mutations are rare but possible in HCM patients. In conclusion, this study highlights the important role of genetic testing as a necessary diagnostic tool for performing a definitive etiological diagnosis of HCM.

## 1. Background

Hypertrophic cardiomyopathy (HCM) is a disease of the heart muscle characterized by left ventricular hypertrophy not solely explained by abnormal loading conditions [1,2]. In 50–60% of cases, the disease is inherited as an autosomal dominant genetic trait caused by mutations in genes encoding sarcomeric proteins [1]. Nevertheless, a subgroup of cases is caused by mutations in non-sarcomeric genes or systemic disorders (i.e., genetic syndromes and storage, infiltrative, metabolic and neuromuscular disorders) causing cardiac hypertrophy [1,2,3]. Rasopathies, including Noonan syndrome (NS) and related disorders, are among the possible and most common causes of HCM in children and young adults. NS is a genetic disorder, inherited as an autosomal dominant trait, characterized by a wide spectrum of symptoms and physical features that can present in variable combinations and severity degrees [4]. The most common features of NS are represented by facial dysmorphisms, such as broad forehead, ptosis and hypertelorism, short stature, and congenital heart diseases (CHDs). The majority of NS patients (about 80–90%) have a cardiovascular involvement that can include a broad range of CHDs and/or early-onset HCM [5]. Several genes in the RAS–MAPK signaling pathway cause NS (or a related condition). The first gene described was PTPN11, followed by SOS1, RAF1, KRAS, BRAF, NRAS, MAP2K1 and RIT1 and, recently, SOS2, LZTR1 and A2ML1 [6]. This condition generally occurs during prenatal development, although milder cases may be identified many years later. Genotype–phenotype studies have shown large inter- and intra-familial variability in the clinical features of NS due to the different mutations in the different NS genes [7]. Moreover, previous studies hypothesized that epigenetic mechanisms may be responsible for the variable clinical expression of NS. It was suggested that non-coding regions, located in introns and far away from the annotated genes, can be involved in the etiology of the disease [8]. Additionally, the presence of multiple genetic hits (i.e., polymorphisms or double mutations), favoring a “threshold effect”, can modify systemic and cardiac phenotypes in patients with NS [9]. Although very rare, the coexistence of sarcomeric and RAS-MAPS genes in adults with HCM and mild NS phenotypes should be ruled out in clinical practice.

## 2. Case Report

The subject (33-year-old female) showed regular psychophysical development. He underwent a cardiological evaluation for the first time at the age of 24 years-old and the echocardiographic evaluation showed a mild increase in left ventricular wall thickness (13 mm). Nine years later, following the occurrence of several episodes of palpitations, the patient accessed to our division (Inherited and Rare Cardiovascular Disease Clinic—University of Campania “Luigi Vanvitelli”, Naples, Italy) and underwent a comprehensive clinical evaluation. The physical examination showed mild facial dysmorphisms (Figure 1), short stature and systolic heart murmur; no other signs of cardiovascular disease were reported. The echocardiographic evaluation showed the presence of asymmetric left ventricular hypertrophy, with a maximum wall thickness of 16 mm at the level of the anterior interventricular septum, and mild mitral regurgitation.

Based on clinical and echocardiographic features, the subject underwent genetic testing (next generation sequencings (NGS)] with a panel of 325 genes, known to be associated with cardiomyopathies) (Appendix A), after giving informed written consent according to the procedure established by the local ethics committee. A blood sample in EDTA was collected from the subject. Genomic DNA was extracted using the Maxwell 16 instrument (Promega, Madison, WI, USA), and DNA quality was assessed using a Nanodrop machine. Molecular testing was carried out by analyzing a panel of target genes through NGS-based procedure. The Alyssa software (Agilent, Santa Clara, CA, USA) was used to perform sequence data analysis. This tool allows the alignment of the sequences to the reference genome in order to obtain a list of genomic variants that can be prioritized using a bioinformatic pipeline in order to highlight pathogenic mutations or potentially pathogenic variants. Genome data processing was performed using a home-made bioinformatic pipeline. The MAF threshold was set to 5% using the Illumina Variant Interpreter Software.

Genetic testing showed a heterozygous mutation of the PTPN11 gene (c.1510A>G, p.Met504Val, NM_002834.3), a pathogenic variant that confirms the clinical suspicion of NS [10], and also a heterozygous mutation of the MYBPC3 gene (c.1790G>A, p.Arg597Gln, NM_000256.3), a likely pathogenic variant associated with HCM [11]. The variants identified by NGS were also validated by Sanger sequencing. Thus, the parents were invited to join the cascade program screening; the genetic testing was restricted to the same mutations found in the proband. The father showed the same mutation in the MYBPC3 gene, while the PTPN11 variant was not identified in the parents and was considered a de novo mutation (Table 1).

All the family members (both parents and brothers) underwent a complete cardiological evaluation. The father showed the presence of HCM, while no signs of cardiovascular disease were evidenced in the other family members. However, after comprehensive genetic counselling, in the absence of systemic and cardiac signs of NS, the parents refused to perform the genetic test on their other children (which were underage). The pedigree of the family is represented in Figure 2.

## 3. Discussion

To our knowledge, this is the first case of a young adult with HCM and NS with mild phenotype expression showing a compound heterozygosity in two genes commonly associated with these pathological disorders (PTPN11 and MYBPC3). The mutation of the MYBPC3 gene (p.Arg597Gln, exon18) was a missense mutation affecting the highly conserved amino acid and the last nucleotide of exon 18, and causing a splicing variation. The mutation is classified as pathogenic/likely pathogenic for HCM in several genetic databases. Recent studies showed that this mutation in HeLa cells causes an inclusion of exon 18 in the transcript and an almost-undetectable 248 bp product [11]. The PTPN11 mutation (p.Met504Val) is classified as pathogenic, compatible with NS [12]. Double or triple mutations in sarcomeric genes are generally associated with a more severe phenotype [13,14]. However, little is known regarding the impact of the association of PTPN11 and sarcomeric gene mutations on disease expression. Here, we report a non-severe, familial case of non-obstructive HCM, associated with mild clinical features of NS, diagnosed by a multidisciplinary team of expert cardiologists and geneticists. Cardiac and non-cardiac red flags (RF) represent important diagnostic markers that permit to suspect a specific cause of cardiomyopathy and a distinction between sarcomeric and non-sarcomeric types [1,3], and genetic testing represents a necessary diagnostic tool for performing a definitive etiological diagnosis [15,16,17,18]. Indeed, the association between HCM and pulmonary stenosis and/or dysmorphisms should always raise suspicion of genetic syndromes, even in adults. This is of clinical relevance, since sarcomeric and non-sarcomeric HCM have different disease expression and prognosis [19]. Patients with NS associated with PTPN11 mutations generally carry a high risk of progressive symptoms and heart failure, particularly if a left ventricular outflow obstruction is present [19]. On the other hand, young patients with sarcomeric HCM may present a high risk of life-threatening arrhythmias [1,17]. Thus, in patients with HCM, the presence of cardiac or non-cardiac RF, even in mild expression, should raise suspicion of a non-sarcomeric cause that could explain the clinical phenotype and whose identification could impact the clinical outcome and patient management. Indeed, when a correct genetic diagnosis is made, through an adequate genetic counselling and a cascade screening program, all affected family members can be diagnosed, and preventive strategies can be implemented. On the other hand, it is also important to consider the possible psychological implications for children, and in the absence of a specific early therapeutic intervention and an evident cardiac or specific phenotype, cascade screening can be postponed according to parents’ will.

## 4. Conclusions

We described a rare case of a young adult with HCM and NS, carrying two disease-causing mutations in MYBPC3 and PTPN11. This study highlights the importance of suspecting non-sarcomeric forms of HCM, even in the presence of a mild phenotype, with a multidisciplinary team of cardiologists and geneticists with specific expertise in inherited and rare diseases.

## Figures and Tables

**Figure 1 genes-11-00947-f001:**
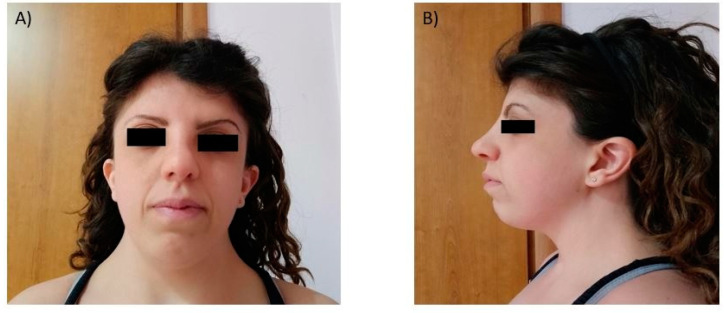
(**A**,**B**). Photographs of the index patient. Features include a wide mouth, sloping forehead, prominent nasal bridge, prominent nose and low attachment of the ears.

**Figure 2 genes-11-00947-f002:**
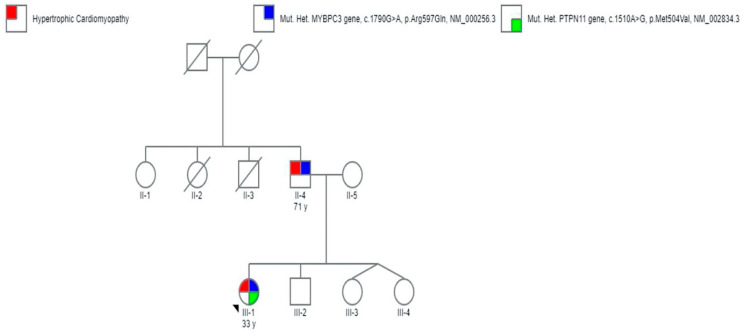
Pedigree of the family of propositus. Genotype and phenotype are defined according to the legend inset. Open symbols represent subjects with a negative genotypes and phenotypes. I-1, I-2, II-2 and II-3 represent subjects with unknown phenotypes.

**Table 1 genes-11-00947-t001:** Clinical and genetic characteristics of the patients. HCM, hypertrophic cardiomyopathy.

ID	Cardiological Features	Genetic Variants	ACMG Score	Gnomad Frequencies	ClinVar Classification
III.1	HCM	PTPN11 gene (c.1510A>G, p.Met504Val, NM_002834.3)	PS3	0.00000882	Pathogenic
		MYBPC3 gene (c.1790G>A, p.Arg597Gln, NM_000256.3)	PM1	0.00000886	Likely pathogenic
II.4	HCM	PTPN11 WT/WT MYBPC3 gene (c.1790G>A, p.Arg597Gln, NM_000256.3)	PM1	0.00000886	Pathogenic
II.5	No evidence of cardiovascular abnormalities	PTPN11 WT/WT MYBPC3 WT/WT	N/A	N/A	N/A

## Data Availability

The data that support the findings of this study are available from the corresponding author upon reasonable request.

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
