# Peer review of "Combined PTPN11 and MYBPC3 Gene Mutations in an Adult Patient with Noonan Syndrome and Hypertrophic Cardiomyopathy"

_genes, 2020, doi:10.3390/genes11080947_

Round 1

Reviewer 1 Report

Authors reported an interesting case report of Noonan syndrome and HCM.

The manuscript is well-written and only some minor should be clarified:

1.- In the pedigree do not state clear if both clinical and genetic assessment were performed in III.2, II.3 and III.4.

Please clarify text.

2.- A table including genetic and clinical can help readers to have a correlation.

Please include ACMG score, gnomAD frequencies, ClinVar classification...

3.- Finally, are there previous publications concerning any of both variants?

Author Response

Reviewer 1

Question n. 1

Authors reported an interesting case report of Noonan syndrome and HCM. The manuscript is well-written and only some minor should be clarified:

1.- In the pedigree do not state clear if both clinical and genetic assessment were performed in III.2, II.3 and III.4.Pleaseclarify text.

Answer n.1: Thank you for your comment. We better clarified in the text that the parents of the proband refused to perform the genetic test on the other sons which were underage (III.2,III.3,III.4) (see page 3, lines 89-91).All the family members (III.2,III.3,III.4) underwent a complete cardiological evaluation. All patients at cardiological assessment don’t have any evidence of structural cardiovascular abnormalities.  Only The father of the proband showed the presence of HCM. We specify in the text: “All the family members (both parents and brothers) underwent a complete cardiological evaluation. The father showed the presence of HCM, while no signs of cardiovascular disease was evidenced in the other family members. However, after a comprehensive genetic counselling, in absence of systemic and cardiac signs of NS, parents refused to perform the genetic test on their other children (which were underage).”

Question n. 2:

2.- A table including genetic and clinical can help readers to have a correlation.

Please include ACMG score, gnomAD frequencies, ClinVarclassification...

Answer n. 2:Thank you for your important comment. According with your suggestion, we included the following table in the text.

ID

Cardiological Features

Genetic Variants

ACMG score

gnomAD frequencies

ClinVar classification

III.1

HCM

PTPN11 gene

(c.1510A>G, p.Met504Val, NM_002834.3)

MYBPC3 gene

(c.1790G>A, p.Arg597Gln, NM_000256.3)

PS3

PM1

0.00000882

0.00000886

Pathogenic

Likely pathogenic

II.4

HCM

PTPN11 WT/WT

MYBPC3 gene (c.1790G>A, p.Arg597Gln, NM_000256.3)

-

PM1

-

0.00000886

-

Pathogenic

II.5

No evidence of cardiovascular abnormalities

PTPN11 WT/WT

MYBPC3 WT/WT

-

-

-

-

-

-

Question n.3:

3.- Finally, are there previous publications concerning any of both variants?

Answer n. 3: Thank you for your comment. There are other publications that prove the pathogenicity of the single variants and we added them in the references.

Reviewer 2 Report

A nice example of the importance of complete genetic testing in the assessment of cardiomyopathies and other inherited cardiovascular diseases

Author Response

Reviewer 2

A nice example of the importance of complete genetic testing in the assessment of cardiomyopathies and other inherited cardiovascular diseases

Answer: We thank very much the Referee.

Reviewer 3 Report

The authors present the report of an adult patient affected by hypertrophic cardiomyopathy and found to be double heterozygous for a PTPN11 and a MYBPC3 pathogenic DNA variant.

Major comments.

  1. ABSTRACT: the authors should report the age of the patient
  2. INTRODUCTION: the authors should include details and references about variable expressivity of NS and the current hypothesis/findings that could explain such phenotypic variability
  3. CASE REPORT: page 2, line 45. Define better “mild increase of left ventricular wall thickness” providing the measurement of the wall thickness.
  4. The authors should add a supplementary table with the list of all the genes included in the NGS panel, as well as their coverage and mean read depth.
  5. The methods of the sequencing should be better described (i.e. sequencing platform, bioinformatics pipeline employed for the variants filtering)
  6. Page 2, line 60: remove “likely”, since the PTPN11 c.1510A>G is a pathogenic variant already reported in ClinVar and already described in literature (also add those references reporting this variant)
  7. Page 2, line 62. Add the references reporting the MYBPC3 variant.

Author Response

Reviewer 3

Question n. 1

  1. ABSTRACT: the authors should report the age of the patient

Answer n. 1: We described the patients’ age in the abstract (page 1, line 19).

Question n. 2

  1. INTRODUCTION: the authors should include details and references about variable expressivity of NS and the currenthypothesis/findingsthatcouldexplainsuchphenotypicvariability

 Answer n. 2:Thank you for your comment. We added in the text (page2, lines 46-52): Moreover, several studies hypothesized that epigenetic mechanisms might be responsible for the variable clinical expression of NS. It was suggested that non-coding regions, located in introns and far away from the annotated genes, can be involved in the etiology of disease. Another hypothesis is represented by the presence of polymorphisms or double mutations that can create an addition effect which can be responsible of a stronger phenotype in a patient with NS.”

Question n. 3

  1. CASE REPORT: page 2, line 45. Define better “mild increase of left ventricular wall thickness” providing the measurement of the wall thickness.

Answer n. 3: Thank you for your suggestion. We provided the measurement of the wall thickness (see page 2, lines 57-58).

Question n. 4

  1. The authors should add a supplementary table with the list of all the genes included in the NGS panel, as well as their coverage and mean read depth.

 Answer n.4:Thank you for your comment. The patient underwent genetic testing in an outer genetic laboratory. The DNA sample was sequenced using NGS sequencing panel of 348 genes (not 202, as previously described). Thus, we modified it in the text, and we apologize for the inaccuracy. We have reported the list of the genes included in the NGS panel in the Supplementary Table 1.

Question n. 5

  1. The methods of the sequencing should be better described (i.e. sequencing platform, bioinformatics pipeline employed for the variants filtering)

Answer n.5: Thank you for the suggestion. We have clarified the procedure at pages 3-4, lines 72-79: “A blood sample in EDTA was collected from subject. Genomic DNA was extracted using the Maxwell 16 instrument (Promega, Madison, Wisconsin, USA)and DNAs quality was assessed by Nanodrop. Molecular testing was carried out by analyzing a panel of target genes through a next generation sequencing-based procedure. The Alyssa software (Agilent, Santa Clara, California, USA) was used to perform sequence data analysis. This tool allows the alignment of the sequences to the reference genome in order to obtain a list of genomic variants that can be prioritized using a customizable pipeline in order to highlight pathogenic mutations or potentially pathogenic variants.”

Question 6-7

  • Page 2, line 60: remove “likely”, since the PTPN11 c.1510A>G is a pathogenic variant already reported in ClinVar and already described in literature (also add those references reporting this variant)
  • Page 2, line 62. Add the references reporting the MYBPC3 variant.

Answer n. 6-7: Thank you for your comment. Done!

Round 2

Reviewer 3 Report

The authors should add, in the methods , more details about the bioinformatic pipeline employed for the variants ' filtering. At least, the authors should declare the filtering MAF.

The authors added a supplementary table with the analysed genes, as requested. But also, the in silico coverage of each gene, as well as their mean read depth (obtained for the here reported case), should be added.

Author Response

Response to reviewers

Reviewer3, Round 2

Question n. 1

The authors should add, in the methods , more details about the bioinformatic pipeline employed for the variants ' filtering. At least, the authors should declare the filtering MAF.

Answer n.1: Thank you for your comment. We better clarified in the text the bioinformatic pipeline employed for the variants ' filtering and the filtering MAF (Line 79-81).

Question n. 2:

The authors added a supplementary table with the analysed genes, as requested. But also, the in silico coverage of each gene, as well as their mean read depth (obtained for the here reported case), should be added.

Answer n. 2:Thank you for your suggestion.According to your suggestion, we have updated the supplementary table.
